# Prevention of Tracheo-Innominate Artery Fistula Formation as a Complication of Tracheostomy: Two Case Reports

**DOI:** 10.3390/children9111603

**Published:** 2022-10-22

**Authors:** Byungsun Yoo, Bongjin Lee, June Dong Park, Seong Keun Kwon, Jae Gun Kwak

**Affiliations:** 1Department of Pediatrics, Seoul National University College of Medicine, Seoul 03080, Korea; 2Department of Otorhinolaryngology-Head and Neck Surgery, Seoul National University College of Medicine, Seoul 03080, Korea; 3Department of Thoracic and Cardiovascular Surgery, Seoul National University College of Medicine, Seoul 03080, Korea

**Keywords:** fistula, innominate artery, muscular dystrophy, omphalocele, trachea, tracheostomy

## Abstract

Tracheo-innominate artery fistula (TIF) is a rare complication of tracheostomy and refers to the formation of a fistula between the trachea and innominate artery. Because TIF is fatal, prevention rather than treatment is very important. Here we report the cases of two high-risk patients who underwent tracheostomy, and in whose cases attempts were made to lower the risk of TIF. In the first patient who developed a chest deformity with Duchenne muscular dystrophy, a tracheostomy was performed with a high-level (cricothyroid level) approach compared with the standard tracheostomy. In the second patient, the thoracic cage was relatively small due to a giant omphalocele, and the risk of a fistula forming was decreased by wrapping the innominate artery with an opened polytetrafluoroethylene vascular graft after resolving crowding of the intrathoracic cavity by total thymectomy. There was no TIF occurrence at the outpatient follow-up in either case. We expect that our approaches may be effective intervention measures for preventing TIF.

## 1. Introduction

Tracheo-innominate artery fistula (TIF) is a communication between the anterior aspect of the trachea and the posterior wall of the innominate artery. TIF is known to occur due to erosion of the tracheal wall by the tracheostomy cannula tip and persistent pulsatile contact with the innominate artery adjacent to the erosion site [1]. TIF incidence is rare, at approximately 0.2%–0.45% occurrence after tracheostomy; however, it is a fatal complication of tracheostomy, wherein untreated massive bleeding can block the entire airway and lead to 100% mortality [2,3].

TIF develops because of persistent pulsatile contact between the tracheal wall and adjacent innominate artery, which leads to erosion of the tracheal wall and fistula formation. For this reason, the presence of chest deformity, where the innominate artery and tracheal wall come into contact with each other, is a risk factor for TIF [4,5,6,7,8].

To date, research on TIF has mainly focused on its management [9,10,11,12,13,14], with studies on its prevention being limited. In a previous document, it was reported that prophylactic brachiocephalic trunk separation was performed as a method to prevent the occurrence of TIF [15]. Another document introduced preventive innominate artery division or ligation [16]. However, these methods were invasive methods that required surgery on large arteries. Therefore, we report two cases in which a relatively less invasive intervention plan for TIF prevention was implemented to patients with a high risk of TIF.

## 2. Case Description

### 2.1. Case 1

A 23-year-old man was admitted with the chief complaint of dyspnea 5 days prior to admission. The patient had a medical history of Duchenne muscular dystrophy (DMD) since age 3 years of age, along with developmental delay, which was diagnosed and treated at another tertiary hospital. The patient had progressive muscle weakness and scoliosis and underwent scoliosis correction surgery 9 years prior and percutaneous endoscopic gastrostomy surgery 5 years prior to admission. He was hospitalized several times for recurrent pneumothorax.

At the time of admission, the patient appeared to be chronically ill. His height and weight were 165 cm and 24 kg, respectively. His vital signs were as follows: blood pressure, 134/101 mmHg; pulse rate, 113 beats/min; respiratory rate, 40 breaths/min; and body temperature, 36.2 °C. Percutaneous oxygen saturation was 98%, with O_2_ at 10 L/min through a partial rebreathing facial mask. The chest wall expanded symmetrically, but the right lung sounds were hoarse and left lung sounds were absent.

The patient was diagnosed with a large left pneumothorax. Even after chest tube insertion, the pneumothorax was not resolved and persisted, and pleurodesis was performed on the 3rd day of hospitalization. Non-invasive ventilation with a facial mask was then applied in response to persistent hypercapnia, but on the 4th day of hospitalization, venous blood gas analysis showed a worsening type 2 respiratory failure with pH 7.02 and carbon dioxide partial pressure 110 mmHg. He was transferred to the pediatric intensive care unit (ICU), where endotracheal intubation and mechanical ventilation were performed. Neck computed tomography (CT) was performed in order to evaluate TIF risk before the tracheostomy showed that the trachea was compressed by the innominate artery, and the risk of TIF was deemed to be high with standard tracheostomy (Figure 1A,B). Therefore, tracheostomy was performed at a higher (cricoid) level than that with standard tracheostomy in order to decrease TIF risk.

Neck CT performed after the tracheostomy showed that the distance between the tracheostomy cannula tip and innominate artery was approximately 30 mm (Figure 2A,B). Tracheostomy was completed without acute complications, and the pneumothorax improved. The patient was transferred to the general ward on the 29th day of hospitalization and was discharged on the 46th day of hospitalization. After approximately 2 years, the patient’s DMD progressed, and he died due to the discontinuation of life-sustaining treatment. However, during his life, the patient had no TIF occurrence.

### 2.2. Case 2

A patient with a giant omphalocele, confirmed by prenatal ultrasonography, was born via a caesarean section at a gestational age of 37 weeks and 1 day, and the birth weight was 2740 g. His Apgar score was 1 point at 1 min and 2 points at 5 min; after resuscitation in the operating room, he was transferred to the neonatal ICU and treatment was continued. He underwent tracheostomy on the 135th day of life due to prolonged endotracheal intubation and mechanical ventilation. Bronchoscopy performed at the time of tracheostomy revealed severe tracheomalacia; thus, a long tracheostomy cannula (uncuffed cannula with an inner diameter of 4.0 mm and length of 6 cm) was laid. After weaning with home mechanical ventilation, the plan was to transfer the patient to the general ward. However, because of repeated weaning failure, the patient was transferred to the pediatric ICU on the 191st day of life. Chest CT showed that the innominate artery was adjacent to the trachea (Figure 3A,C,D), and the left main bronchus was pressed between the left pulmonary artery and the descending thoracic artery, showing luminal narrowing (Figure 3B).

Surgery was planned to resolve the narrowing of the left main bronchus and to reduce the TIF risk. At 201 days of age, total thymectomy and pulmonary arteriopexy (the transition zone of the main pulmonary artery and left pulmonary artery was fixed anteriorly towards the sternum) were performed under upper partial sternotomy (Figure 4A). Then, we longitudinally opened a piece (about 7–8 mm in length) of 10 mm–polytetrafluoroethylene (PTFE) vascular graft and wrapped the innominate artery with this vascular graft without re-closure (Figure 4B). We also added a couple of fixation sutures between the graft and the adventitia of the artery to prevent migration of this vascular graft.

In this manner, direct contact between the innominate artery and the trachea could be avoided. In chest CT it was confirmed after the surgery that the distance between the innominate artery and trachea was longer than that before, that the PTFE vascular graft was well positioned around the innominate artery (Figure 5A), and that the left main bronchus narrowing was resolved (Figure 5B).

Subsequently, the patient was weaned off with home mechanical ventilation, transferred to the general ward, and discharged on the 331st day of life. He was followed up on an outpatient basis, and no respiratory difficulties or other complications were noted.

## 3. Discussion

We incorporated methods to minimize the risk of TIF in the tracheostomy procedures in the above two cases. Although the first patient eventually died, it was not due to tracheostomy complications, such as TIF. The cause of death in this patient was his worsening underlying disease, but TIF did not occur for approximately 2 years before patient’s death. In the second case, TIF did not occur during the outpatient follow-up, and we believe that this was attributable to the increased distance between the innominate artery and trachea on follow-up CT.

In the first case, as a method to lower the risk of TIF occurrence, we first considered sternal resection, innominate artery reimplantation, and standard tracheostomy. This can relieve the carina from the innominate artery and secure airway patency. However, the patient had DMD and showed severe cachexia. Therefore, the soft tissue around the sternum would be insufficient to cover the defect after sternal resection. A small defect volume may be covered by the pectoralis major muscle flap, whereas a large defect volume should be covered by the vertical rectus abdominis myocutaneous flap. As an alternative to the above method, we considered tracheostomy with a high-level (cricothyroid) approach. With this method, surgical wound healing is difficult, subglottic stenosis progresses, and above all, vocal cord injury may make vocalization difficult. As mentioned earlier, this patient had DMD as an underlying disease, and the life expectancy of patients with this disease is approximately 20–25 years [17,18]. Considering the life expectancy of the patient, risk of infection, and difficulty of wound healing, tracheostomy at the cricothyroid level is a reasonable option.

Unlike the first case, the second case did not undergo CT scan for TIF risk assessment before tracheostomy. CT performed to determine the cause of repeated ventilation weaning failure confirmed narrowing of the left main bronchus and a high risk of TIF. Therefore, to reduce the risk of TIF occurrence, we considered two options. First, this patient had a long tracheostomy cannula adjacent to the innominate artery due to tracheomalacia. Therefore, if posterior tracheopexy was performed for tracheomalacia and a relatively short conventional cannula used instead of a long cannula, the risk of TIF occurrence would be reduced. The second option was to remove the thymus to secure space in the intrathoracic cavity to relieve tracheal compression by the innominate artery. If the second option alone failed to solve the problem, an additional incision and lateral thoracotomy would have been required for the first option. Fortunately, sufficient free space was secured with thymectomy, and only additional graft placement and anterior pulmonary arteriopexy needed to be performed.

Many literature reviews and reported cases introduce the treatment of patients with already-ruptured TIF. Failure in primary bleeding control via first aid maneuver such as digital compression or ballooned double-lumen endobronchial tube led to fatality [10,12]. A few studies have shown successful treatment through vascular intervention via endovascular stent-graft repair [13,14]. Some have repaired the innominate artery and made a flap between the trachea and artery as a mechanical buffer [5,19,20], while others simply ligated the innominate artery to remove the cause of TIF [9,21,22,23]. Few authors have suggested preventive interventions to reduce the risks of TIF. One converted the tracheostomy tube into a length-adjustable type of device in order to avoid the mechanical contact [15], while others preventively ligated the innominate artery [16,24,25,26]. Due to the high mortality associated with TIF, dealing with patients with a ruptured TIF is quite challenging. Considering that ligation or division of the innominate artery is associated with the risks of brain ischemia [27], ligating the innominate artery as a preventive measure is questionable. A less invasive method such as high tracheostomy, or surgical methods that maintain normal vessel anatomy, as in our cases, could be novel alternatives for TIF prevention.

## 4. Conclusions

Because TIF is extremely fatal, its prevention is better than treatment after it occurs. Therefore, we introduced the high-level approach, thymectomy, and graft placement to lower the risk of TIF, and evaluated whether these methods effectively lowered the risk of TIF occurrence. Since each method is not without complications (especially the high-level approach), decisions should be made considering the overall condition and risk–benefit assessment for the patient. In addition, long-term monitoring for TIF occurrence after the surgery is required.

## Figures and Tables

**Figure 1 children-09-01603-f001:**
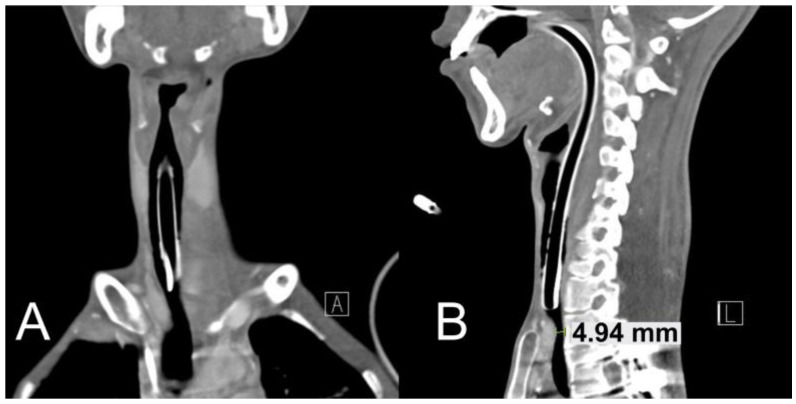
Neck computed tomography images before tracheostomy. (**A**) The coronal view shows that the tip of endotracheal tube and innominate artery are adjacent. (**B**) The sagittal view shows the innominate artery pressing the trachea.

**Figure 2 children-09-01603-f002:**
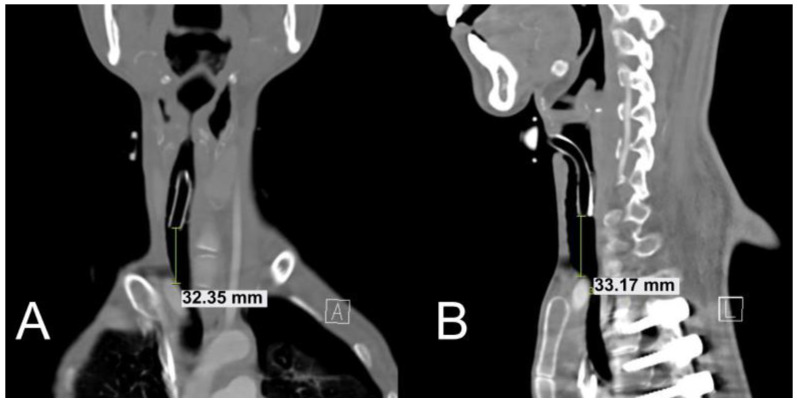
Neck computed tomography images after the high-level (cricothyroid level) tracheostomy approach. In (**A**) coronal and (**B**) sagittal views, the distance between the tracheostomy cannula tip and innominate artery was approximately 33 mm.

**Figure 3 children-09-01603-f003:**
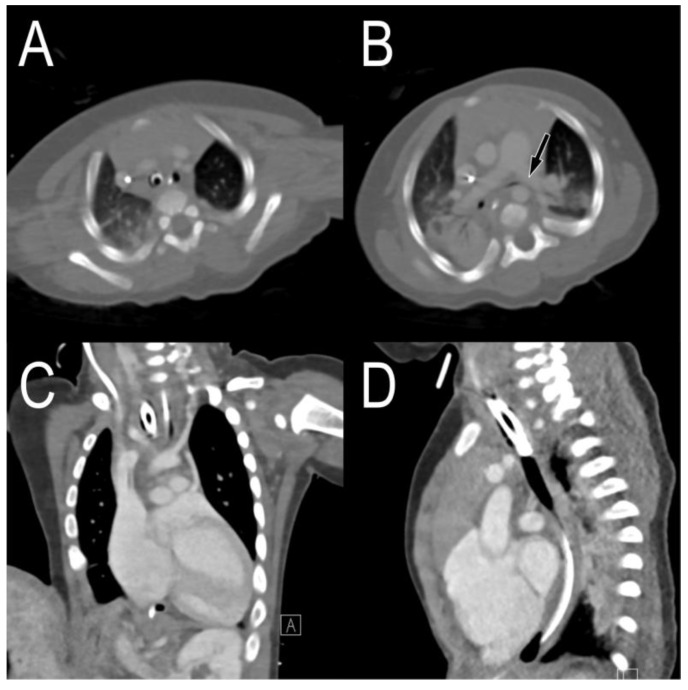
Chest computed tomography images before tracheostomy. (**A**) The axial view shows that the trachea and innominate artery are adjacent, and (**B**) the left main bronchus is compressed by the left pulmonary artery. In (**C**) coronal and (**D**) sagittal views, the cannula tip is attached to the innominate artery.

**Figure 4 children-09-01603-f004:**
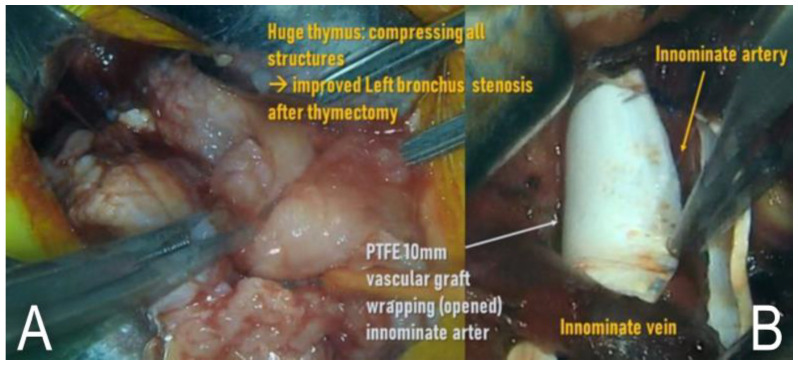
Surgical filed photo images. (**A**) Huge thymus was removed, and (**B**) polytetrafluoroethylene vascular graft wrapped the innominate artery.

**Figure 5 children-09-01603-f005:**
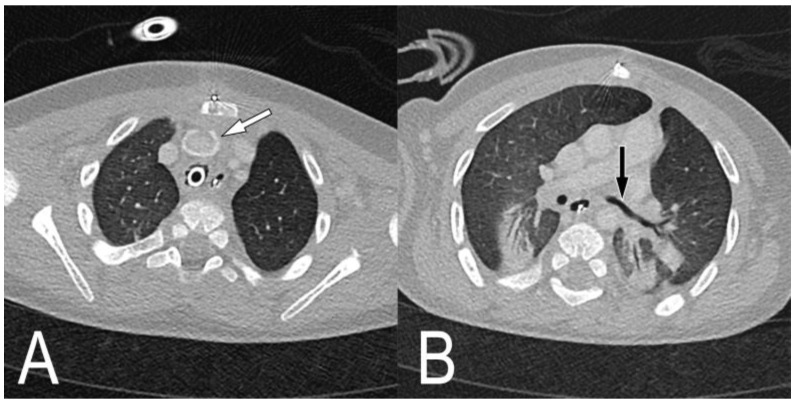
Neck computed tomography images after tracheostomy. (**A**) The white arrow shows that the polytetrafluoroethylene vascular graft surrounds the innominate artery, and that there is a gap between the innominate artery and trachea. (**B**) The narrowing of the left main bronchus is resolved after anterior-pulmonary arteriopexy (black arrow).

## Data Availability

Not applicable.

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
