# Peer review of "Prevention of Tracheo-Innominate Artery Fistula Formation as a Complication of Tracheostomy: Two Case Reports"

_children, 2022, doi:10.3390/children9111603_

Round 1
Reviewer 1 Report
At first, I would like to congratulate the authors for the successful management of both the cases. Tracheo-innominate artery fistula is a fatal complication. The authors have reduced the risk of TIF by doing a high tracheostomy (first case) and wrapping the innominate artery in a vascular graft (second case) to avoid any direct contact between the artery and tracheal wall.
I have some comments:
-The first case is an adult (23-year old). Although the second case is a newborn, it is better to submit it in a Journal that has a broader readership.
-Do you perform CT scan in all cases to assess the risk of TIF before proceeding to surgery?
-I would advise the authors to do a review a literature and highlight the measures/steps taken by other surgeons to minimize the risk of TIF. Please mention this in one paragraph/table.
Author Response
At first, I would like to congratulate the authors for the successful management of both the cases. Tracheo-innominate artery fistula is a fatal complication. The authors have reduced the risk of TIF by doing a high tracheostomy (first case) and wrapping the innominate artery in a vascular graft (second case) to avoid any direct contact between the artery and tracheal wall.
→ Thank you for your encouraging comments. We have answered each of the points mentioned below. Thank you.
I have some comments:
-The first case is an adult (23-year old). Although the second case is a newborn, it is better to submit it in a Journal that has a broader readership.
→ As you said, the first case was that of an adult patient. In our hospital, if there is an underlying disease that was diagnosed as a child and was followed up and treated into adulthood, treatment is continued at the children’s hospital in adult patients, not the adult hospital. Since the above patient had a congenital disease of Duchenne muscular dystrophy that was diagnosed and followed up into the adult years, pediatric intensivists and pediatric otolaryngologists were involved in the treated, therefore that patient was treated in the PICU. Your comment has allowed us to realize that reporting adult cases in the ‘Children’ journal might be confusing to some readers. However, like our hospital, there are many hospitals where treatment continues in children’s hospitals even if the patients have reached adulthood, and we think this case has meaning to them. Furthermore, it expands the spectrum of readers, especially for physicians treating adult patients with congenital diseases.
-Do you perform CT scan in all cases to assess the risk of TIF before proceeding to surgery?
→ Not all preoperative CT scans are available for all patients undergoing tracheostomy. As in the second case, if a tracheostomy was performed in the NICU, there may be cases in which a CT scan is not performed prior to surgery. Also, in case of emergency tracheostomy, prior CT examination may not be performed. However, since a patient died from TIF several years ago, patients undergoing elective tracheostomy have routinely performed contrast CT before surgery at our center. Thank you for your valuable comment.
-I would advise the authors to do a review a literature and highlight the measures/steps taken by other surgeons to minimize the risk of TIF. Please mention this in one paragraph/table.
→ Thanks for the important comment. We have followed your recommendations to review the methods that have been tried to reduce the risk of developing TIF and related to TIF. We revised the manuscript to include this content in a separate paragraph in the ‘Discussion’ section. We think that you made a very important point, which allowed our manuscript to be more refined. Thank you.

Reviewer 2 Report
This paper presents two cases where authors have sought measures to prevent tracheo-arterial fistulas. As the authors have stated, tracheo-innominate arterial fistulas (TIF) are very rare occurring only in 0.2 – 0.45% after tracheostomy. Authors present 2 cases where prevention methods were used.
Comments:
Authors state in introduction that studies on prevention methods are limited. I would expect that the previously reported methods for prevention would be introduced in introduction or in discussion.
Case 1 is not really a pediatric patient, which appears a bit strange as the manuscript is submitted to pediatric journal under pediatric surgery section.
Figure 1 (before tracheostomy) mentions tracheostomy tube. Should be endotracheal tube as the patient was intubated, right?
Case 2 had two separate tracheostomy operations and the second one was performed due to severe tracheomalacia. Therefore, thymectomy and aortopexy seem appropriate, as well as cautious approach with innominate artery. Are there any other publications with this technique to avoid TIF?
My main concern is that we cannot know whether the selected approaches actually reduced the risk of TIF, as the risk (even in patients with higher risk) is considerably low.
Author Response
This paper presents two cases where authors have sought measures to prevent tracheo-arterial fistulas. As the authors have stated, tracheo-innominate arterial fistulas (TIF) are very rare occurring only in 0.2 – 0.45% after tracheostomy. Authors present 2 cases where prevention methods were used.
Comments:
Authors state in introduction that studies on prevention methods are limited. I would expect that the previously reported methods for prevention would be introduced in introduction or in discussion.
→ We appreciate your valuable comments. Although there are many reports on TIF treatment, there are only a few studies on the prevention of TIF. We completely agree with your opinion that it is necessary to mention them in the introduction. Therefore, we added related contents to the Introduction and Discussion sections.
Case 1 is not really a pediatric patient, which appears a bit strange as the manuscript is submitted to pediatric journal under pediatric surgery section.
→ As you said, the first case was that of an adult patient. In our hospital, if there is an underlying disease that was diagnosed as a child and was followed up and treated beyond childhood and adolescent years, treatment was continued at the children’s hospital even as an adult. Since the above patient had a congenital disease of Duchenne muscular dystrophy which was diagnosed when the patient was a child, pediatric intensivists and pediatric otolaryngologists treated him at the PICU. Your comment has led us to realize that the report of adult cases in the ‘Children’ journal might be confusing to some readers. However, like our hospital, there may be hospitals where treatment continues in children’s hospitals even if they have just reached adulthood, and we think this case has meaning to them. Furthermore, we believe it expands the spectrum of readers to physicians that care for adult patients with congenital diseases.
Figure 1 (before tracheostomy) mentions tracheostomy tube. Should be endotracheal tube as the patient was intubated, right?
→ Thank you for the correction. The manuscript was revised as recommended.
Case 2 had two separate tracheostomy operations and the second one was performed due to severe tracheomalacia. Therefore, thymectomy and aortopexy seem appropriate, as well as cautious approach with innominate artery. Are there any other publications with this technique to avoid TIF?
→ When reviewing the previously reported literature, previous studies have dealt with treatment after the onset of TIF. In one document, there was a case in which the innominate artery was repaired and a flap was placed between the trachea and the artery during TIF treatment. It may be a similar method to our second case, but the difference is that we did not flap during the repair of blood vessels after TIF, rather prevented TIF by covering the artery with a graft beforehand. In addition, there were reports that a length-adjustable type of cannula was placed as a preventive method and that the innominate artery was ligated. To the best of our knowledge, there has been no literature stating that the method we reported was used. This is explained in detail in the Discussion section of the main text. Thank you.
My main concern is that we cannot know whether the selected approaches actually reduced the risk of TIF, as the risk (even in patients with higher risk) is considerably low.
→ We deeply agree with your concern. It would be tempting to say that the risk of TIF was reduced due to the preventive measures that were applied, especially from the perspective of those that applied the intervention, however, it is difficult to prove it scientifically because the actual probability of occurrence is very low. However, the medical staff who treat patients will be willing to do anything if there is a way that is expected to help the patient’s safety. In that sense, we would appreciate your understanding about the importance of our cases and reporting the methods we tried.

Round 2
Reviewer 1 Report
The authors have modified the manuscript as per my comments. In the revised version, the overall scientific quality of the manuscript has improved significantly. Although the paper has some limitations, it provides a great addition to the current literature.
Reviewer 2 Report
No further comments.